# Comparative Volatilomic Profile of Three Finger Lime (*Citrus australasica*) Cultivars Based on Chemometrics Analysis of HS-SPME/GC–MS Data

**DOI:** 10.3390/molecules27227846

**Published:** 2022-11-14

**Authors:** Rosaria Cozzolino, José S. Câmara, Livia Malorni, Giuseppe Amato, Ciro Cannavacciuolo, Milena Masullo, Sonia Piacente

**Affiliations:** 1Institute of Food Science, National Research Council (CNR), Via Roma 64, 83100 Avellino, Italy; 2CQM—Centro de Química da Madeira, Universidade da Madeira, Campus da Penteada, 9020-105 Funchal, Portugal; 3Departamento de Química, Faculdade de Ciências Exatas e da Engenharia, Universidade da Madeira, Campus da Penteada, 9020-105 Funchal, Portugal; 4Dipartimento di Farmacia, Via Giovanni Paolo II n.132, 84084 Fisciano (Salerno), Italy

**Keywords:** finger limes, volatile composition, HS-SPME/GC–MS, authentication, statistical analysis, discrimination

## Abstract

Finger lime is receiving growing attention as an ingredient of gastronomic preparations of haute cuisine for its delicious flavor and fragrance and for its appealing aspect. Volatile compounds play a crucial role in determining the organoleptic characteristics of the fruit and its pleasantness for consumers. The aim of the present study was to investigate the volatile profiles by headspace solid phase micro-extraction (HS-SPME) coupled to gas chromatography–mass spectrometry (GC–MS) in the peel and, for the first time, in the pulp of three Australian finger lime cultivars grown in Sicily (southern Italy): Pink Pearl, Sanguinea, and Faustrime, allowing to overall identify 84 volatile organic compounds (VOCs). The analytical data showed that the three cultivars were characterized by distinct volatile chemotypes: limonene/sabinene/bicyclogermacrene in the Pink Pearl, limonene/γ-terpinene/bicyclogermacrene in the Sanguinea, and limonene/β-phellandrene/γ-terpinene in the Faustrime. Moreover, some volatiles, found exclusively in one cultivar, could be considered potential markers of the individual cultivar. PCA allowed us to clearly discriminate the samples into three clusters: the first related to the Sanguinea peel, the second to the Faustrime peel, and a third group associated with the Pink Pearl peel along with the pulp of the three cultivars. Accordingly, the VOCs that mostly contributed to the differentiation of the three finger lime cultivars were also identified. Among them, D-limonene, sabinene γ-terpinene, α-pinene, α-phellandrene, β-myrcene, p-cymene, linalool, δ-elemene, ledene, bicyclogermacrene β-citronellol, α-bergamotene, α-caryophillene, and β-bisabolene, have been previously reported to exhibit important biological activities, suggesting that these cultivars, in addition to possessing unique volatile profiles, can show promise for several applications in pharmaceutical and food industry, namely for development of functional foods.

## 1. Introduction

Finger lime (*Citrus australasica* F. Muell.) is one of six *Citrus* species endemic to Australia. Its native range is mainly restricted to the rainforests of Queensland and New South Wales [1].

Compared with other citrus fruit, finger lime is probably one of the most attractive and interesting, especially for its high levels of natural genetic diversity. The tree can reach up to 6 m in height and the fruit presents a finger-like shape, characterized by a wide range of colors of pulp and peel, including purple, green, yellow, and pink) [1]. Finger lime is also called caviar lime, as one of the most distinctive traits of the fruit is the spherical pulp sacs which give the pulp a caviar-like appearance (Figure 1) [1,2].

Some native fruits were selected to be commercially cultivated only in the 1990s and afterward, some Australian endemic varieties were considered for commercial production [3]. Finger lime can adapt to the unusual soil and different climate conditions found in Australia, from extreme drought to rainforest [3]. As a result, they have many agronomical traits of potential interest, including tolerance to drought, reduced fruiting period, resistance to diseases, and a unique genetic diversity [3]. Because of its appealing colors and unique organoleptic and flavor characteristics, in Australia, finger limes are generally eaten fresh in fruit salads, while the pulp is also used in gourmet food to garnish seafood (oyster, caviar, and sushi), or added in alcoholic drinks. Currently, global interest is increasing in finger lime and cultivar (CV) breeding programs and/or trial plantations have been developed both in the United States and China [1]. Interest has also been garnered in the use of this fruit in industrial applications [3]. The content of complex polyphenols contributes to the variability of colors among the varieties and the antioxidant properties reported for the finger lime species. For instance, the investigation of four CVs in Florida highlights the occurrence of flavonoids, anthocyanins, vitamin C, citric acid, and sugars characterized by antioxidant capacity at higher levels in the peel compared to the pulp in the four selections [4]. The occurrence of phenolic compounds is related to potential health-promoting effects supporting the nutraceutical use of fruits. Finger lime extracts inhibited in vitro the NO-releasing and the inflammation-related cytokines and alleviated LPS-induced upregulation of several patterns [2]. In general, *Citrus* species have a very peculiar and attractive aroma, which makes them particularly agreeable. Finger lime aroma and flavor are highly variable. However, the peel of this fruit is frequently defined as having an intense floral lime aroma, while the juice as being sour and relatively strongly pungent [3].

Odor perception is strictly related to the presence of volatile organic compounds (VOCs). To date, analyses of the volatiles in the peel oil of commercially-grown Australian finger limes are limited [1]. Trozzi et al. (1993) described that the volatile content of a cold-pressed oil of “Faustrime”, a trigeneric hybrid of *M. australasica* × *Fortunella* sp. × *C. aurantifolia*, cultivated in Italy, was dominated by limonene (43.2%), citronellal (16.3%) γ-terpinene (4.5%), and α-phellandrene (4.5%) [5]. The VOCs composition of the essential oil of the peel of *C. australasica* var. Sanguinea adapted in Sicily was evaluated by Ruberto et al. (2000) by a GC–MS analysis which allowed to identify, as principal constituents, bicyclogermacrene (26%), α-pinene (10%), spathulenol (10%), and *cis*-β-ocimene (5.1%), whereas limonene was detected as only 1.2% of the total oil [6]. A few years later, Lota et al. (2002) compared the VOCs pattern of the peel and leaf oil of 43 lemons and limes produced in Corsica, showing that the lime species could be clustered into three chemotypes: limonene, limonene/β-pinene, and limonene/β-pinene/γ-terpinene. In this study, the peel oil of C. *australasica* presented a unique profile among the citrus fruit, with sabinene (19.6%) as the second most abundant compound after limonene (51.1%) [7]. Delort and Jaquier (2009) and Delort et al. (2015) investigated the VOCs profile of peel samples from three Australian-grown finger lime varieties by GC–MS and GC–FID analysis, identifying three chemotypes never observed in any other *Citrus* species, including limonene/sabinene, limonene/citronellal/isomenthone, and limonene/citronellal/citronellol [8,9]. The comparative analysis performed by Delort and Jaquier (2009) and Delort et al. (2015) also identified some volatile constituents which were detected for the first time in a *Citrus* extract and some VOCs which tended to be specific to one finger lime CV [8,9]. Very recently, Johnson et al. (2022) determined the total phenolic compounds, the antioxidant capacity the total anthocyanin content, the ascorbic acid, and the amino acids citrulline and arginine content of five Australian-grown finger lime CVs. The VOCs profile was performed by GC–MS, with a total of 113 volatile compounds identified in the peel extracts. The predominant volatile compounds identified in the finger lime CVs included limonene, γ-terpinene, β-citronellol, and citronellal. As a result of its high ascorbic acid content, moderate phenolic and flavonoid content, and a unique VOCs profile responsible for its characteristic organoleptic properties, the Australian finger lime was thought to have the potential for commercial development as a functional food [1]. Generally, the compounds present at lower concentrations are more important in determining the organoleptic profile compared to those in high amounts. Moreover, the contents of these VOCs, called odor-active volatiles, are sometimes so low as to be hard to detect [10]. In the last decades, the introduction of the headspace solid phase micro-extraction (HS-SPME), coupled with gas chromatography–mass spectrometry (GC–MS), has allowed for a more accurate investigation of the aromatic profile in food. This solvent-free extraction technique is quite simple and does not require any kind of sample pre-treatment, thus avoiding any loss of VOCs reflecting the true native volatile pattern of the citrus fruit without any modification [10].

The aim of the present study was to characterize and compare, both from a qualitative and semi-quantitative point of view, the VOCs profile of peel and, for the first time, of the pulp of three finger lime CVs, namely Pink Pearl, Sanguinea, and Faustrime, grown in the Sicily (southern Italy), by HS-SPME/GC–MS methodology, as these metabolites play a crucial role in the organoleptic sensation of the fruit and consumers’ acceptability.

## 2. Results and Discussion

### 2.1. Volatile Constituents in the Peel and Juice

The volatile profiles obtained from the three-finger lime samples present different VOCs profiles, as shown in Appendix A, which display typical total ion chromatograms (TIC) of peel and juice of the CV Pink Pearl, Sanguinea, and Faustrime, respectively. A total of 84 volatile compounds were identified by HS-SPME/GC–MS in the peel and the juice of the three different finger lime CVs, including sesquiterpenes (33), monoterpenes hydrocarbons (19), oxygenated monoterpenes (12), aldehydes (7), esters (4), alcohols (4), and others (5), as listed in Appendix A, which also reports the VOCs abbreviation code, the experimental and literature reported Kovats index, and the identification method (Appendix A). Among all volatile components, 54 have been already identified in the peel extract of different finger lime CVs [1,3,5,6,7,8,9,11,12,13], while a further 30 VOCs, which seem not been previously observed in any finger lime CV, have been reported in the volatile patterns of various *Citrus* species [10,14,15,16,17,18,19]. Previous studies concerning the VOCs profiles in different finger lime CVs have highlighted that, based on a comparative qualitative and quantitative analysis of the volatile pattern in the peel extract, this fruit can be clustered into diverse chemotypes, which are not only unique in the lime species but also the genus *Citrus* [3,5,6,7,8,9]. The comparative analysis also demonstrated the presence of volatile compounds which have never been detected in any other citrus and of some volatile constituents specific to one CV [3]. The number of volatile compounds in the finger lime peels ranged from 74 to 46 for the Pink Pearl and the Faustrime, respectively, while in the juice they ranged from 45 to 41 for the Sanguinea and the Pink Pearl, respectively (Table 1). According to the literature on *Citrus* species, in all CVs, the number and the content of the VOCs in the peels were higher than in the juices and distinct volatile chemotypes for the three finger lime varieties of the present study were observed (Table 1) [1,9,16]. Figure 2A reports the volatile content overall separately detected in the pulp and the peel in all the three investigated finger lime CVs (inner annulus), while the external annulus refers to the individual contribution, in terms of the total volatile pattern that each finger lime CV has separately given to the peel and the pulp. On the other hand, Figure 2B indicates that monoterpene hydrocarbons, followed by sesquiterpenes, are clearly the most abundant VOCs identified in all fruit samples.

As expected, limonene was the main volatile of the total VOCs content both in the finger lime peels (65.7 to 29.3%) and in the juices (66.8 to 32.2%) (Table 1) [1]. The dominant chemotype of the Pink Pearl peel was limonene/sabinene/bicyclogermacrene (63.2/9.5/7.2%) (Table 1). Other volatiles identified in moderately high concentrations (>1%) included α-thujene (2.4%), δ-elemene (2.2%), γ-terpinene (2.0%), β-myrcene (1.6%), terpinen-4-ol (1.4%), and ledene (1.1%) (Table 1). On the other hand, the principal volatile components in the juice of the Pink Pearl CV were limonene (32.2%), followed by γ-terpinene (15.8%) and terpinen-4-ol (15.2%). Other compounds present at a concentration >1% were sabinene (7.3%), α-terpinene (6.6%), α-thujene (4.6%), α-terpinolene (3.9%), β-phellandrene (2.5%), cis-β-ocimene (2.0%), p-cymene (1.6%), ledene (1.5%), β-myrcene (1.2%), and bicyclogermacrene (1.1%) (Table 1). Moreover, minor compounds, including 2-hexenal (0.2%), 1-hexanol (0.1%), 3-hexen-1-ol (0.2%), cis-sabinene hydrate (0.3%), trans-sabinene hydrate (0.2%), citronellal (0.3%), carvone (0.1%), β-citronellol (0.4%), α-gurjunene (0.1%), calarene (0.1%), epizonarene (0.2%), α-caryophillene (0.3%), α-farnesene (0.1%), α-muurolene (0.1%), sphatulenol (0.1%), and tetradecene (0.1%) were detected only in the peel, while aromadendrene (0.2%) was identified only in the juice (Table 1). Notably, ethyl acetate (E1), hexyl butyrate (E4), 2-butenal (Ald1), 2-hexen-1-ol (Al4), carvone (MO6), β-bourbonene (Sesq5), and epiglobulol (Sesq31) were detected only in the peel of Pink Pearl CV while trans-sabinene hydrate was observed both in the peel and in the juice, suggesting that these VOCs can be considered markers of this CV (Table 1).

The chemotype of the Sanguinea peel was characterized by limonene/γ-terpinene/bicyclogermacrene (65.7/8.8/7.0%), with four other constituents detected at concentrations between 3.4 and 1.4%, including sabinene, ledene, δ-elemene, and β-myrcene (Table 1). These results are in contrast with the VOCs composition of the essential oil of the peel of the CV Sanguinea investigated by Ruberto et al. (2000), who reported that the main volatiles were bicyclogermacrene (26%), α-pinene (10%), spathulenol (10%), and cis-β-ocimene (5.1%), while limonene represented only 1.2% of the total oil [6]. This discrepancy can be presumably ascribed to different factors, including the row material (peel vs. essential oil), the different pedo-climatic environment, and the diverse techniques used for the volatile’s extraction. Limonene (66.8%), γ-terpinene (5.5%), and ledene (4.1%) were the most abundant components in the juice of CV the Sanguinea (Table 1). Other VOCs found at a concentration >1% were aromadendrene (3.8%), bicyclogermacrene (3.5%), terpinen-4-ol (1.7%), β-myrcene (1.4%), cis-β-ocimene (1.3%), and α-terpinolene (1.2%) (Table 1). Furthermore, cis-3-hexanal (0.1%), 2-hexenal (0.5%), citronellal (0.2%), linalool (0.3%), β-citronellol (0.1%), epibicyclosesquiphellandrene (0.1%), β-guaiene (0.1%), β-bisabolene (0.1%) tetradecene (0.1%), and pentadecane (0.1%) were detected only in the peel. On the contrary, α-phellandrene, α-copaene, α-muurolene, and sphatulenol, all at a percentage of 0.1%, were found only in the juice (Table 1). Remarkably, β-elemene (Sesq9) has been observed only in the VOCs profile of the Sanguinea peel, proposing that this sesquiterpene could be indicated as a potential marker for this variety.

The chemotype of the Faustrime peel can be classified as limonene/β-phellandrene/γ-terpinene (29.3/21.1/9.5%) (Table 1). The same three volatiles were observed at the highest amounts also in the juice with the following percentages: limonene at 35.7%, followed by β-phellandrene at 23.5% and γ-terpinene at 12.6% (Table 1). In the Faustrime peel, VOCs detected at a concentration >1% included p-cymene (6.3%), β-myrcene (4.5%), α-phellandrene (4.4%), β-bisabolene (4.0%), β-citronellol (2.3%), bicyclogermacrene (2.1%), α-pinene (2.0%), α-caryophillene (2.0%), linalool (1.7%), sabinene (1.5%), α-thujene (1.2%), and α-terpinolene (1.1%) (Table 1). Conversely, in the juice, in addition to α-pinene, α-phellandrene, β-myrcene, p-cymene, β-citronellol, and α-caryophillene, which were also present in the peel, four other constituents were found with a content >1%, including α-bergamotene, terpinen-4-ol, and α-terpinene (Table 1). Moreover, minor compounds, including 2-hexenal (0.5%), aromadendrene (0.4%), citronellal (0.3%), trans-carveol (0.2%), hexen-1-ol propionate (0.1%), cis-sabinene hydrate (0.1%), cis-limonene oxide (0.1%), and cis-carveol (0.1%) were identified only in the peel, while ledene (0.1%) was found only in the juice (Table 1).

Finally, cis-limonene oxide and nerolidol were exclusively found in the peel of the Faustrime CV (Table 1). These molecules, which seem to be specific to one CV, could be considered volatile markers of this CV.

It is worth mentioning that, as there is no authentic chemical standard for bicyclogermacrene, this compound has been putatively identified, based on its high percentage of similarity (>90%) and according to previous reports about the characterization of the peels of other finger lime CVs [1,6].

### 2.2. Targeted Multivariate Statistical Analysis of Secondary Metabolites

Data obtained by HS-SPME/GC–MS analyses were submitted to multivariate data analysis. Specifically, the Principal Component Analysis (PCA) was performed to discriminate the peel and fruit extracts of *C. australasica* CV Sanguinea, CV Pink Pearl, and CV Faustrime [20]. For the unsupervised PCA, a data matrix was generated by reporting the different samples (peel and fruit) of the three CVs (six observations) of *C. australasica* and the RPA% of the metabolites (variables) identified by the HS-SPME/GC–MS analysis. The choice of principal components was established based on the fitting (R2X) and predictive (Q2X) values of the PCA. The resulting model, obtained after scaling data by Pareto scaling, showed good fitness and the absence of outliers. PC1 contributed to 84.4% of the variance followed by PC2, which contributed to 8.8%. Hence, the first two PCs exhibited a total variance of 93.2%. Therefore, the analyzed varieties were well discriminated. The targeted PCA score plot showed three different clusters (Figure 3A), while the PCA loading plot highlighted the signals responsible for the distribution on the PCA score plot (Figure 3B) [21].

Figure 3A highlights a clear distinction between clusters relating to the peel of the Sanguinea (on the right part of the top quadrant) and the Faustrime (on the left part of the bottom quadrant) CVs according to the second main component (PC2). Data related to the juice of the three CVs along with the skin of the Pink Pearl are concentrated in a single cluster of the PCA model (Figure 3A). The PCA loading plot (Figure 3B) shows the markers responsible for the distribution of the three groups (Figure 3A). Metabolites in the loading plot that are distant from the origin can be considered chemical markers of the variety as a confirmation of their different distribution in different clusters. In particular, the cluster is related to the peel of var. The Sanguinea is characterized by a higher content of the hydrocarbon monoterpenes sabinene and γ-terpinene (MH5 and MH13, respectively), and by the sesquiterpenes δ-elemene, ledene, and bicyclogermacrene (Sesq3, Sesq18, and Sesq22, respectively). Volatile monoterpenes and sesquiterpenes play multiple roles in plant responses and are synthesized by various types of terpene synthases (TPSs) [22]. *Citrus* species are characterized by large volumes of essential oils, most components of which are volatile terpenoids that are responsible for numerous and crucial biological activities of valuable applications in human health [22]. Many studies in fact have documented that the antimicrobial, antioxidant, anti-inflammatory, and anticancer activities of essential oils are directly correlated with the quality and quantity of their chemical constituents, elucidating their mechanism of action and phytotherapeutic targets [22]. In particular, M13, generally reported among the predominant volatiles identified in different finger lime CVs [1], is the second most abundant component characterizing the chemotype of the Sanguinea peel (Table 1). MH5, MH13, Sesq3, Sesq18, and Sesq22, all described with woody, citrus, and herbal notes, show several important pharmaceutical properties, most of all antimicrobial and anti-inflammatory abilities [14,23,24,25]. Specifically, several assays, including DPPH and ABTS tests, have shown that essential oils with γ-terpinene as among the main volatile components exhibited appreciable antioxidant and antiradical activity [10]. Moreover, reactions among γ-terpinene, as a pure compound with ABTS•+ and DPPH•, implied that this terpene can directly scavenge radicals as well as inhibit DNA and erythrocytes from oxidation [10]. On the other hand, Cucho-Medrano et al. (2021) have demonstrated that the different medicinal properties of the *Croton* genus might be related to the high content of bicyclogermacrene (Sesq22), which has been detected among the dominant constituent of the VOCs fingerprint of the different species [26]. The skins of the Faustrime CV are differentiated by the higher presence of hydrocarbon monoterpenes α-pinene, α-phellandrene, β-myrcene, and p-cymene, (MH1, MH7, MH8, and MH15, respectively), along with the oxygenated monoterpenes linalool and β-citronellol (MO3 and MO7, respectively), and the sesquiterpenes α-bergamotene, α-caryophillene, and β-bisabolene, (Sesq8, Sesq16, and Sesq23, respectively). The hydrocarbon monoterpene fraction (MH1, MH7, MH8, and MH15), all described by a terpenic, citrus, slightly green flavor, and the two oxygenate monoterpenes (MO3 and MO7), both with citrus and floral odor, have been often found in moderate concentrations in lime species [1,9,12]. Moreover, these terpenes have been reported to extend different biological activities [1,12,23]. In particular, Spyridopoulou et al. (2017) have demonstrated that mastic oil, which contains about 67.7% α-pinene and 18.8% myrcene, is able to induce a statistically significant anti-tumor effect on colon cancer in mice, but not α-pinene, myrcene, or a combination thereof, providing evidence that the interaction of different monoterpenes in a plant matrix can produce a remarkable synergistic or additive effect which seems to enhance the biological properties [27]. Regarding the three sesquiterpenes, Sesq8, Sesq16, and Sesq23 are typically detected in lime species and are responsible for the woody lime character perceived in the fresh peel of this fruit [1,9]. Additionally, α-caryophillene (Sesq16) and α-bergamotene (Sesq8), among the major components in the bergamotene oil, have been found to possess marked anti-inflammatory actions, while a combination of α-bergamotene (Sesq8), β-bisabolene (Sesq23), and α and β-selinene, the four major components of *Copaifera reticulata*, have been reported to have good activity against oral pathogens [28]. Data related to the juice of all three CVs and the peel of the Pink Pearl CV form a single cluster of the PCA model (Figure 3A). These samples are directly correlated to limonene and β-phellandrene (MH10 and MH11, respectively), which occurred in higher content in their VOCs profiles (Table 1, Figure 3B). All the other identified compounds lay in proximity to the origin, so they can not represent characteristic markers of discrimination.

Limonene, which greatly contributes to the fruity scent owing to its low odor threshold, has been reported to possess several healthy properties [11]. Specifically, various in vitro assays have demonstrated that limonene extends a concentration-dependent decrease in free radical formation and, when evaluated together with other terpenoids, it has resulted to be among the most powerful scavengers of the free radical DPPH [10]. Singh et al. (2010) have observed that the antifungal, antiaflatoxigenic, and antioxidant activities of the essential oils of *Citrus maxima* and citrus sinensis are almost due to limonene, the major component in both oils. Moreover, authors reported that limonene showed an even better antiaflatoxigenic efficacy at half of the concentration (250 ppm) that requested for both essential oils and their 1:1 combination to have the same effect [29]. Consequently, *Citrus* spp. essential oils and/or limonene alone could be recommended as a plant-based antimicrobial, as well as useful additives for food preservation to prolong shelf life and improve the quality of stored food commodities, without modifying their organoleptic properties [29]. On the other hand, limonene has been revealed to exhibit other potential biological activities, including hypocholesterolemic and anticarcinogenic properties [30].

The essential oil of *Stachys lavandulifolia* Vahl. subsp. lavandulifolia (Lamiaceae), rich in β-phellandrene (27%), has been reported to extend antimicrobial activity against Staphylococcus aureus and Salmonella typhimurium and high DPPH radical scavenging action [31]. In general, these biological properties are generally observed in essential oils characterized by the presence of a high proportion of MH11 [26].

## 3. Materials and Methods

### 3.1. Chemicals and Materials

All chemicals, standards, and reagents were from Sigma-Aldrich (St. Louis, MO, USA). Ultrapure water (18 M Ω cm at 25 °C) was obtained from a Millipore Milli-Q purification system (Millipore Corp., Bedford, MA, USA). The HS-SPME fibers (divinylbenzene/carboxen/polydimethylsiloxane-DBV/CAR/PDMS, with 50/30 μm film thickness and 1 cm fiber length), and the glass vials were from Supelco (Bellofonte, PA, USA). Helium, at a purity of 99.999% (Rivoira, Milan, Italy), was used as the carrier gas in the GC system. The capillary GC column HP-Innowax (30m × 0.25 mm × 0.5 μm) was from Agilent J&W (Agilent Technologies Inc., Santa Clara, CA, USA).

### 3.2. Plant Material

The three varieties of *C. australasica* Mill. (1 kg for each variety) were purchased by an agronomic producer in Sicily, a region of southern Italy, in September 2021. A voucher specimen is deposited in the Department of Pharmacy of the University of Salerno. The peel and juice of the fruits were manually separated and immediately stored at −80 °C until analysis.

### 3.3. Volatile Organic Compounds (VOCs) Analysis

#### 3.3.1. Sample Preparation and HS-SPME Procedure

The HS-SPME extraction procedure was carried out according to Figueira et al. (2020) with some variations [14]. In order to obtain a representative sample from each CV of *C. australasica* Mill., samples to analyze were obtained as a pooled sample from 1 kg of fruit. Briefly, 5 mL of finger lime juice, placed into a 20 mL vial with a PTFE-coated silicone septum, and was added to 0.5 g of NaCl and 5 µL of 3-octanol (0.4 µg/mL), used as the internal standard (IS). For the peels, 250 mg of a thinly sliced sample were introduced into a 20 mL HS vial, and 5 µL of 3-octanol (0.4 µg/mL) was added as IS. Samples were equilibrated for 20 min at 45 °C prior to analysis. For the sampling, a fiber coated with 50/30 µm DVB/CAR/PDMS was used. HS-SPME extraction was performed by exposing the fiber to the equilibrated samples’ headspace for 20 min at 45 °C to adsorb the VOCs. Successively, the fiber was automatically introduced into the GC injector port at 250 °C for 10 min for the desorption of the extracted VOCs. At their first use, SPME fibers were conditioned as advised by the producer, but below the maximum suggested temperature. The fibers were daily conditioned at 250 °C for 10 min into the GC injector port and the blank samples were carried out. All the analyses were performed in triplicate Three individual pooled samples of the juice and peel were separately extracted by HS-SPME in triplicate and each extract was also analyzed by GC–MS in triplicate.

#### 3.3.2. Gas Chromatography-Mass Spectrometry Analysis (GC–MS)

The chromatographic separation of the volatile compounds from juices and peels was performed using a gas chromatographer, model GC 7890A, equipped with a capillary column HP-Innowax and coupled to a mass spectrometer 5975 C (system from Agilent Technologies, Santa Clara, CA, USA). The oven temperature was initially held at 40 °C for 1 min and then ramped to 220 °C at 2.5 °C/min and kept for 10 min, for a total GC run time of 83 min. Helium was used as the carrier gas at a flow rate of 1 mL/min. The injection port operated in the splitless mode at 250 °C. The temperatures of the transfer line, the quadrupole, and the ionization source were set at 270, 150, and 230 °C, respectively. The electron impact (EI) mass spectra were acquired at 70 eV and the mass range was 30–300 m/z. The electron multiplier was established to the auto-tune mode and the ionization current was 10 µA. VOCs were identified through a comparison of their mass spectra with those listed in the NIST05/Wiley07 libraries database. Furthermore, the Linear Retention Indices (LRIs) were calculated using a series of n-alkanes (C8-C22) and compared with the available retention data reported in the literature for the polar column (www.pherobase.com, www.flavornet.org, www.ChemSpider.com, https://pubchem.ncbi.nlm.nih.gov, webbook.nist.gov; accessed on 1 July 2022). Identifications were also confirmed by comparison of the retention times of the chromatographic peaks with those, when available, of commercial standards analyzed under the same conditions. Compounds that were not identified using an authentic chemical standard were considered putatively identified. For individual volatiles, the peak area was calculated from the total ion chromatogram (TIC) and semi-quantified by relative comparison with the peak area of the IS (Relative Peak Area, RPA%), according to Figueira et al. (2020) [14].

### 3.4. Multivariate Data Analysis

Principal Component Analysis (PCA) was performed. The data obtained by HS-SPME/GC–MS analysis were processed using SIMCA P+ software 12.0 (Umetrix AB, Umea Sweden). The data matrix was generated by reporting peels and juices of three CVs (six observations) of *C. australasica*, and the relative peak areas of the identified volatile metabolites (eighty-four variables). Pareto scaling was applied to all data from the matrix.

## 4. Conclusions

In this study, HS-SPME-GC–MS was used to characterize the VOCs profile in the peel and, for the first time, in the juice of three Australian finger lime CVs grown in Sicily (southern Italy), namely Pink Pearl, Sanguinea, and Faustrime. In total, 84 VOCs were identified. The analytical data obtained showed that the three CVs are characterized by distinct volatile chemotypes with unusual ratios of major volatiles: limonene/sabinene/bicyclogermacrene in CV Pink Pearl, limonene/γ-terpinene/bicyclogermacrene in CV Sanguinea and limonene/β-phellandrene/γ-terpinene in CV Faustrime. Moreover, some constituents were exclusively found in one CV, (e.g., ethyl acetate (E1), hexyl butyrate (E4), 2-butenal (Ald1), 2-hexen-1-ol (Al4), carvone (MO6), β-bourbonene (Sesq5) epiglobulol (Sesq31), *trans*-sabinene hydrate in the Pink Pearl, β-elemene (Sesq9) in Sanguinea, and cis-limonene oxide and nerolidol in the peel of Faustrime), suggesting that these metabolites can be considered markers of individual CV. The uniqueness of the VOCs profile of the three finger limes was also shown by PCA, which highlighted a clear discrimination of the samples in three clusters The first correlated to the peel of the Sanguinea, the second to the peel of Faustrime, and a third group formed by the peel of Pink Pearl along with the juice of the three varieties. According to the PCA results, the cluster related to the Sanguinea peel is characterized by a higher content of the two hydrocarbon monoterpenes (MH5 and MH13) and three sesquiterpenes (Sesq3, Sesq18, and Sesq22). The peel of Faustrime is differentiated by the higher presence of four hydrocarbon monoterpenes (MH1, MH7, MH8, and MH15), along with two oxygenate monoterpenes (MO3 and MO7) and the three sesquiterpenes (Sesq8, Sesq16, and Sesq23). Data related to the juice of all three varieties and the peel of the Pink Pearl showed that these samples are directly correlated to limonene and β-phellandrene (MH10 and MH11).

Among the VOCs that mostly contributed to the differentiation of the three finger lime varieties, limonene, sabinene γ-terpinene, α-pinene, α-phellandrene, β-myrcene, p-cymene, linalool, β-citronellol, α-caryophillene, and the putatively identified δ-elemene, ledene, bicyclogermacrene, α-bergamotene, and β-bisabolene have been previously reported to exhibit important biological activities, suggesting that these Australian fruits acclimatized in Sicily, in addition to possessing distinctive volatile compositions, show promise for commercial development as functional foods.

## Figures and Tables

**Figure 1 molecules-27-07846-f001:**
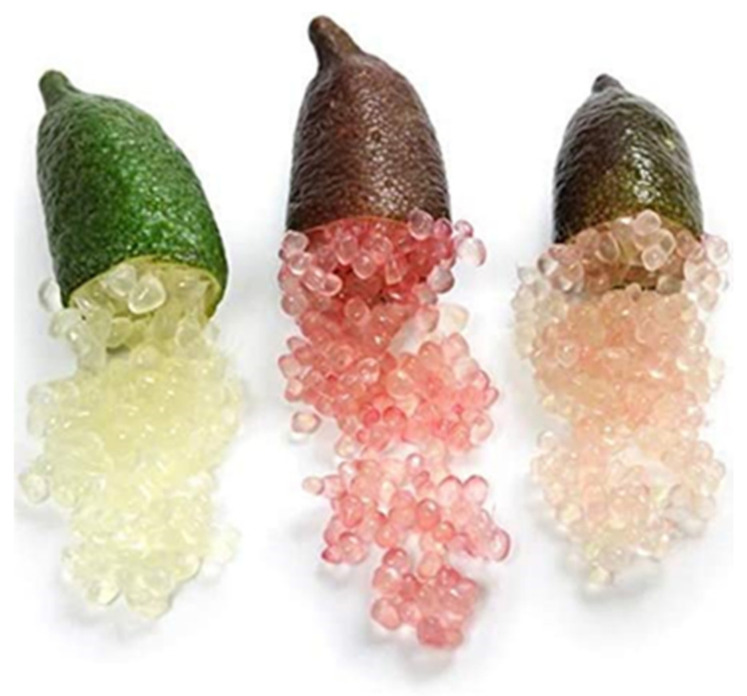
Typical finger lime fruit.

**Figure 2 molecules-27-07846-f002:**
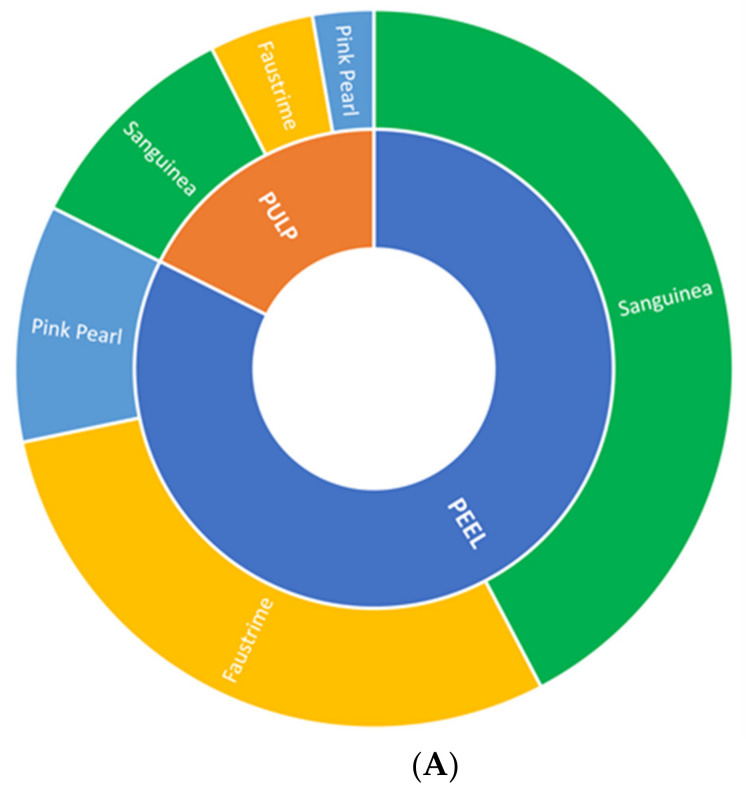
(**A**) Total volatile fraction between the juice and peel in all the three investigated finger lime cultivars (inner annulus) and total volatile content separately contained in the peel and the pulp (external annulus) in each investigated finger lime cultivar; (**B**) Distribution of the VOCs identified in the juice and the peel of the three investigated finger lime cultivars by chemical classes. MH: Monoterpene hydrocarbons; Sesqui: Sesquiterpenes; E. Esters; Ald: Aldehydes; Al: Alcohols; MO: monoterpene alcohols; O: others.

**Figure 3 molecules-27-07846-f003:**
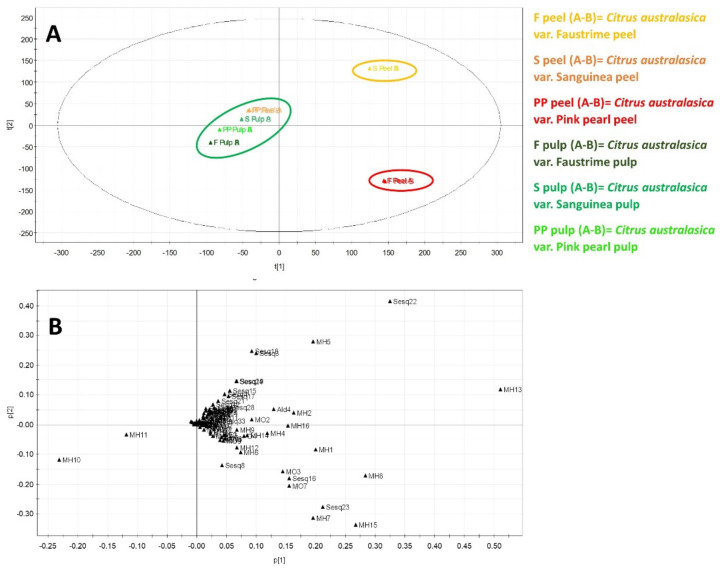
Principal component analysis of specialized metabolites in *C. australasica* varieties obtained by GC–MS targeted analysis: (**A**) PCA score scatter plot; (**B**) PCA loading plot.

**Table 1 molecules-27-07846-t001:** VOCs identified in all *C. australasica* Mill. varieties.

VOCs	Codes	Peel			Juice			Variance P	
		Pink Pearl	Sanguinea	Faustrime	Pink Pearl	Sanguinea	Faustrime		
**Esters**									
Ethyl acetate	E1	1.1 a	0.0 b	0.0 b	0.0 b	0.0 b	0.0 b	0.1	*
*cis*-3-Hexen-1-ol acetate	E2	3.4 b	4.5 c	8.1 d	0.0 a	0.0 a	0.0 a	9.8	**
Hexen-1-ol propionate	E3	1.0 b	27.7 c	78.9 d	0.0 a	0.0 a	0.0 a	923.2	**
Hexyl butyrate	E4	4.5 c	0.0 b	0.0 a	0.0 a	0.0 a	0.0 a	3.0	*
**Aldehydes**									
2-Butenal	Ald1	2.2 b	0.0 a	0.0 a	0.0 a	0.0 a	0.0 a	0.7	*
Hexanal	Ald2	15.2 d	24.3 e	31.2 f	3.1 a	5.8 b	4.1 c	125.3	**
*cis*-3-Hexanal	Ald3	5.1 b	54.0 c	0.0 a	0.0 a	0.0 a	0.0 a	428.7	*
2-Hexenal	Ald4	48.9 b	494.4 d	302.7 c	0.0 a	0.0 a	0.0 a	39,849.0	*
Octanal	Ald5	0.0 a	18.9 c	83.3 d	0.0 a	0.0 a	14.8 a	951.1	*
2-Heptenal	Ald6	2.5 c	0.0 a	0.0 a	0.0 a	0.0 a	1.8 b	1.2	*
Nonanal	Ald7	0.0 a	15.1 c	28.9 d	0.0 a	0.0 a	4.5 b	125.8	**
**Alcohols**									
2-Penten-1-ol	Al1	2.9 b	25.3 d	10.5 c	0.0 a	0.0 a	0.0 a	92.2	***
1-Hexanol	Al2	11.9 c	0.0 a	0.0 a	0.0 a	0.0 a	0.8 b	21.1	***
3-Hexen-1-ol	Al3	41.2 d	0.0 a	0.0 a	0.0 a	5.0 c	2.1 b	243.1	**
2-Hexen-1-ol	Al4	3.4 b	0.0 a	0.0 a	0.0 a	0.0 a	0.0 a	1.7	***
**Monoterpenes hydrocarbons**									
α-Pinene	MH1	72.8 b	717.0 e	1252.2 f	23.6 a	208.9 d	112.1 c	214,826.9	*
α-Thujene	MH2	549.1 d	864.3 f	792.6 e	273.3 c	123.8 b	36.2 a	110,621.3	***
Camphene	MH3	2.9 d	9.5 e	12.7 f	0.7 b	0.0 a	2.0 c	24.7	*
β-Pinene	MH4	48.7 a	276.9 b	383.9 c	20.3 a	36.0 a	28.3 a	22,722.2	*
Sabinene	MH5	2209.4 e	3079.5 f	937.0 d	437.5 c	203.7 b	19.0 a	1,377,296.9	*
δ-3-Carene	MH6	15.9 a	53.8 d	337.0 f	49.4 c	58.0 e	27.0 b	13,538.8	****
α-Phellandrene	MH7	31.6 a	36.9 b	2821.8 d	38.1 b	29.2 a	509.4 c	1,131,513.3	****
β-Myrcene	MH8	363.2 d	1254.1 e	2888.0 f	74.3 a	315.6 c	256.2 b	1,053,379.6	*
α-Terpinene	MH9	220.2 c	335.1 d	408.3 f	392.5 e	163.4 b	98.4 a	14,852.4	****
Limonene	MH10	14,634.6 a	60,239.4 a	18,686.7 a	1919.4 b	14,700.3 a	3659.2 c	2,161,236.8	****
β-Phellandrene	MH11	155.8 c	429.2 d	13,435.1 a	151.6 c	119.6 b	2408.2 e	776,303.6	*
*cis*-β-Ocimene	MH12	65.8 a	151.4 d	419.1 f	120.6 c	292.4 e	90.0 b	17,233.2	*
γ-Terpinene	MH13	453.4 a	8061.7 f	6044.8 e	941.8 b	1200.9 c	1290.0 d	9,413,482.5	*
*trans*-β-Ocimene	MH14	45.4 c	150.9 e	129.4 f	29.5 a	87.8 d	33.3 b	7339.2	*
p-Cymene	MH15	24.9 a	372.1 c	4026.6 d	93.9 a	129.4 b	316.7 c	2,250,635.0	*
α-Terpinolene	MH16	149.6 b	666.2 e	709.2 f	229.9 c	260.3 d	95.6 a	64,890.5	*
Allocimene	MH17	1.3 a	5.3 d	28.6 f	2.0 c	8.9 e	1.7 b	100.5	*
*cis*-Sabinene hydrate	MH18	62.9 d	33.1 b	35.3 c	0.0 a	0.0 a	0.0 a	622.8	*
*trans*-Sabinene hydrate	MH19	35.2 b	0.0 a	0.0 a	1.6 a	0.0 a	0.0 a	185.6	*
**Oxygenated Monoterpenes**									
cis-Limonene oxide	MO1	0.0 a	0.0 a	44.8 b	0.0 a	0.0 a	0.0 a	304.6	***
Citronellal	MO2	77.9 c	215.3 e	186.4 d	0.0 a	0.0 a	3.4 b	8764.6	****
Linalool	MO3	9.9 c	256.6 d	1081.6 f	1.2 a	8.4 b	264.3 e	157,690.6	***
Isopulegol	MO4	0.0 a	21.5 c	0.0 a	0.0 a	0.0 a	4.0 b	69.4	****
Terpinen-4-ol	MO5	324.3 b	576.7 e	440.6 d	906.3 f	382.2 c	126.8 a	63,507.7	**
Carvone	MO6	12.9 b	0.0 a	0.0 a	0.0 a	0.0 a	0.0 a	25.0	**
β-Citronellol	MO7	98.0 b	106.2 c	1457.4 e	1.4 a	4.5 a	158.3 d	293,500.9	**
*cis*-p-Mentha-1(7),8-dien-2-ol	MO8	2.6 b	0.0 a	30.5 c	0.0 a	0.0 a	0.0 a	137.4	**
*trans*-Carveol	MO9	5.6 c	7.5 d	106.5 e	0.0 a	0.0 a	1.7 b	1631.0	**
Benzenmethanol	MO10	2.4 c	0.0 a	0.0 a	0.0 a	0.0 a	1.5 b	1.0	**
*cis*-Carveol	MO11	2.4 c	2.8 d	87.6 e	0.0 a	0.0 a	1.2 b	1129.4	**
Methyleugenol	MO12	4.2 c	0.0 a	11.3 d	0.0 a	0.0 a	1.2 b	18.2	*
**Sesquiterpenes**									
α-Cubebene	Sesq1	10.9 b	15.9 c	0.0 a	0.0 a	0.0 a	0.0 a	45.5	**
α-Copaene	Sesq2	0.0 a	44.3 c	0.0 a	0.0 a	14.5 b	0.0 a	290.6	**
δ-Elemene	Sesq3	510.2 d	1362.2 e	0.0 a	24.9 b	380.3 c	0.0 a	253,917.8	*
Bicycloelemene	Sesq4	85.1 d	256.8 e	0.0 a	5.3 b	48.4 c	0.0 a	8972.8	*
β-Bourbonene	Sesq5	6.7 b	0.0 a	0.0 a	0.0 a	0.0 a	0.0 a	6.8	*
α-Gurjunene	Sesq6	15.2 c	62.7 e	0.0 a	2.4 b	24.4 d	0.0 a	534.3	*
Aristolene	Sesq7	8.8 c	87.4 e	0.0 a	1.9 b	18.2 d	0.0 a	1053.2	*
α-Bergamotene	Sesq8	0.0 a	0.0 a	444.5 c	0.0 a	0.0 a	329.5 b	37,511.5	**
β-Elemene	Sesq9	0.0 a	21.3 b	0.0 a	0.0 a	0.0 a	0.0 a	68.5	*
Calarene	Sesq10	13.3 c	58.7 e	0.0 a	1.8 b	19.0 d	0.0 a	463.6	*
Aromadendrene	Sesq11	0.0 a	274.0 d	245.0 c	9.7 b	843.8 e	0.0 a	96,958.4	**
Epizonarene	Sesq12	51.1 c	105.9 d	0.0 a	0.0 a	24.2 b	0.0 a	1627.7	**
γ-Gurjunene	Sesq13	7.1 b	47.6 d	0.0 a	0.0 a	11.4 c	0.0 a	313.1	**
Epibicyclosesquiphellandrene	Sesq14	4.8 b	46.4 c	0.0 a	0.0 a	0.0 a	0.0 a	315.9	**
Valencene	Sesq15	52.6 c	336.6 e	0.0 a	4.9 b	85.5 d	0.0 a	15,467.2	***
α-Caryophillene	Sesq16	77.3 c	188.3 e	1293.6 f	0.0 a	38.0 b	155.6 d	223,355.3	*
β-Guaiene	Sesq17	44.7 c	259.4 d	0.0 a	9.5 b	0.0 a	0.0 a	9635.1	*
Ledene	Sesq18	249.6 d	1579.2 f	0.0 a	88.8 c	904.4 e	29.1 b	368,786.8	*
Germacrene D	Sesq19	158.4 c	527.9 d	0.0 a	8.5 a	103.1 b	0.0 a	38,052.9	**
β-Selinene	Sesq20	34.3 c	69.1 e	0.0 a	3.3 b	46.4 d	0.0 a	756.5	**
α-Selinene	Sesq21	30.3 c	158.5 e	0.0 a	3.3 b	53.2 d	0.0 a	3424.2	**
Bicyclogermacrene	Sesq22	1663.5 e	6457.0 f	1349.6 d	63.3 b	776.6 c	28.8 a	5,280,975.4	***
β-Bisabolene	Sesq23	0.0 a	56.8 e	2565.5 f	1.3 b	5.8 c	9.5 d	986,299.6	**
δ-Cadinene	Sesq24	139.6 d	524.5 e	0.0 a	8.6 b	128.6 c	5.0 a	36,889.0	**
α-Farnesene	Sesq25	28.4 d	48.9 e	0.0 a	2.6 b	16.9 c	0.0 a	350.7	***
Cadina 1,4 diene	Sesq26	7.1 b	19.4 c	0.0 a	1.1 a	7.3 b	0.0 a	51.4	**
α-Muurolene	Sesq27	16.3 d	30.2 e	0.0 a	1.0 b	13.6 c	0.0 a	134.7	**
Germacrene B	Sesq28	76.7 e	161.9 f	61.0 d	8.5 a	44.6 c	11.2 b	2895.8	**
Calamenene	Sesq29	2.6 b	6.0 d	0.0 a	0.0 a	4.6 c	0.0 a	6.3	**
α-Calacorene	Sesq30	2.1 b	2.9 c	0.0 a	0.0 a	0.0 a	0.0 a	1.6	**
Epiglobulol	Sesq31	1.2 b	0.0 a	0.0 a	0.0 a	0.0 a	0.0 a	0.2	**
Nerolidol	Sesq32	0.0 a	0.0 a	1.5 b	0.0 a	0.0 a	0.0 a	0.3	*
Sphatulenol	Sesq33	16.1 d	40.0 f	31.8 e	1.8 b	13.6 c	0.6 a	229.2	
**Others**									
2-Ethyl-furan	O1	0.5 b	3.8 d	1.7 c	0.0 a	0.0 a	0.0 a	2.1	**
Tridecane	O2	2.1 b	29.2 d	84.3 e	0.0 a	0.0 a	12.6 c	979.4	***
Tetradecene	O3	12.4 c	51.2 d	0.0 a	0.0 a	0.0 a	3.5 b	370.8	*
Tetradecane	O4	5.1 b	21.9 d	107.3 e	0.0 a	0.0 a	9.7 c	1573.2	***
Pentadecane	O5	7.2 b	109.5 d	195.9 e	0.0 a	0.0 a	28.9 c	5783.8	***

Mean values of three samples. For each compound, the mean values followed by a different letter are significantly different (*p* < 0.05) according to the Student–Newman–Keuls (SNK) test (*p*-value * <0.1; ** <0.01; *** <0.001; **** <0.0001. Concentrations were calculated as RPA (%).

## Data Availability

Not applicable.

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
