# Peer review of "Comparative Volatilomic Profile of Three Finger Lime (Citrus australasica) Cultivars Based on Chemometrics Analysis of HS-SPME/GC–MS Data"

_molecules, 2022, doi:10.3390/molecules27227846_

Round 1

Reviewer 1 Report

This paper describes a well conducted study to characterize qualitatively and quantitatively the VOCs profile of peel and, for the first time, of the pulp of three finger lime cultivars cultured in Southern Italy. Since these metabolites are very important for the taste and other organoleptic properties of fruits the scope of the research is interesting and reasonable innovative. On the other hand, the authors employed headspace SPME-GC/MS, which is typical as analytical method, however they included PCA in order to document the evaluation of the profiles and possible differences among cultivars and sample matrices. Unfortunately only three cultivars were included in this investigation, though it would be much more interesting to elaborate in more varieties. Anyway, the whole research targets are sound and the cited references are closely related to the subject. The authors provide detailed profiles of total volatile fraction to reveal differences between the juice and the peel in the three investigated finger lime cultivars. Their analytical results are quite interesting regarding the three cultivars’ different volatile chemotypes while some volatiles were found exclusively in a specific cultivar and they may be used as potential markers for this cultivar. Eventually, all important VOCs that help to discriminate the three types of finger limes were successfully identified. I recommend acceptance of the paper, however some effort is needed to improve some places in the text, e.g.:

Although it is cieted that “Figure 2A reports the profile of total volatile fraction among the juice and peel in the three investigated finger lime cvs” further explanation about the meaning of the figure is required, either in the text or in the legend.

Figure 3 definitely requires higher resolution and clarity.

Author Response

Please read the attached file.

Reviewer 2 Report

Section 2.1

The authors need for provide details in the Materials and Methods section of the methods used to obtain any experimental or calculated result especially regarding the values listed in Table1.

Section 2.2 

The authors refer to the PCA that they use as “supervised” but PCA is an unsupervised technique. Did the authors use some type of modified PCA? Please make it clear in the Materials and Methods section.

PCA was used to discriminate between peel and juice extracts. Did the authors expect the VOC signature of the peel and juice extracts to be similar? I would expect to see same sample types in such types of data analysis approaches (e.g. compare peel extract from different cvs or juice extracts from different cvs). What is reasoning behind this approach?

Figure 3 resolution is too low. Please use higher resolution.

Considering the minimum reporting standards suggested by the Metabolomics Standards Initiative (10.1007/s11306-007-0082-2), I would recommend to the authors to provide additional spectral evidence (or at least spectral matching data) for the identifications of the major compounds that were assigned to chemotypes.

Also, according to Table S1 there was no authentic chemical standard used to identify bicyclogermacrene so the identification for this compound is putative. Additionally, the ratio aRI/bRI for this compound is low, at least in comparison to the rest of the reported ratios. Since the authors assign this compound to a chemotype I would expect the authors to provide further experimental evidence to make the identification unambiguous. If this is not possible – I would suggest to the authors to explicitly cite other studies (maybe 2-3) where putatively identified compounds are assigned to such chemotypes and explicitly report in the Results and Discussion section and in the Conclusion that the identification for this compound is putative.

Section 3

All methods used for obtaining any experimental result must be explicitly and clearly described in the Materials and Methods section.

Line 325 replace “suddenly” with “immediately”

Sections 3.3.1.

Was a pooled sample prepared from the 1 kg of fruits? If this is the case, please state this explicitly.

From the text it is assumed that the juice and peel of each variety was extracted and analysed three times. Is this the case? Please provide as much information as possible to make your replication approach clearer E.g. “each juice and peel pooled sample was extracted in triplicate and analysed”.

Section 3.3.2 

Compounds that were not identified using an authentic chemical standard are considered putatively identified and should be reported as such (please see Metabolomics. 2007 Sep; 3(3): 211–221) - https://doi.org/10.1007/s11306-007-0082-2). Please make the necessary text amendments in-line with the reporting guidelines.

Line 356. Please list all the literature sources that were used for performing identifications (RIs, spectra etc).

Line 360 Please add a reference to at least one study (the most relevant to your sample type) that uses the same semi-quantification approach.

Section 3.4.

Please describe in section 3.4 all methods used for data analysis. For Table 1, which values are included and how are the values of each column calculated. E.g. you mentioned p-values of the SNK test in Table 1 but this is not described in the Materials and Methods section. 

Please state in-text (not in the section title) which multivariate statistical technique was used. Did the authors use PCA (unsupervised) or some type of modified PCA? In case the latter was used, please add an explanation in the text as to the motivation for this choice along with at least two references using the same approach in relevant studies.

It is not explicitly clear how the data matrix was prepared; please rewrite this section. I would assume 6 observations per cultivar, three for the juice and three for the peel of each cultivar. This should be made clearer. Did the authors include both sample types (peel and juice) in the same dataset? Usually, PCA is used for analysing the same sample types as one would expect major differences in case of different sample types anyway. Please add some references (relevant to your field) that follow this approach of mixing sample types.

The authors state in this section that analyses were performed in duplicate. Which analyses do they refer to?

Author Response

Please read the attached file.

Round 2

Reviewer 2 Report

Authors have responded to my comments and provided valid rebuttals (where applicable).

After a quick vendor search, bicyclogermene does not appear readily available as a commercial standard, hence, it is probably difficult for the authors to make an unambiguous identification. Regardless, this has to be reported and the authors have provided relevant comments in lines 219-222.

Some final comments below:

Line 226: Still reads that PCA is unsupervised, this needs to be corrected.

Line 152: Define SNK test in the Materials and Methods (Section 3.4)

Section 3.4: PCA must be mentioned in-text, not in the title.
